# Highly Valuable Fish Oil: Formation Process, Enrichment, Subsequent Utilization, and Storage of Eicosapentaenoic Acid Ethyl Esters

**DOI:** 10.3390/molecules28020672

**Published:** 2023-01-09

**Authors:** Mengyuan Yi, Yue You, Yiren Zhang, Gangcheng Wu, Emad Karrar, Le Zhang, Hui Zhang, Qingzhe Jin, Xingguo Wang

**Affiliations:** 1State Key Laboratory of Food Science and Technology, School of Food Science and Technology, National Engineering Research Center for Functional Food, International Joint Research Laboratory for Lipid Nutrition and Safety, Collaborative Innovation Center of Food Safety and Quality Control in Jiangsu Province, Jiangnan University, Wuxi 214122, China; 2Wuxi Children’s Hospital, Children’s Hospital Affiliated to Jiangnan University, Wuxi 214023, China

**Keywords:** ethanolysis, EPA-EE, EPA, extraction, enrichment, concentration, bioavailability, oxidation stability, high purity

## Abstract

In recent years, as the demand for precision nutrition is continuously increasing, scientific studies have shown that high-purity eicosapentaenoic acid ethyl ester (EPA-EE) functions more efficiently than mixed omega-3 polyunsaturated fatty acid preparations in diseases such as hyperlipidemia, heart disease, major depression, and heart disease; therefore, the market demand for EPA-EE is growing by the day. In this paper, we attempt to review EPA-EE from a whole-manufacturing-chain perspective. First, the extraction, refining, and ethanolysis processes (fish oil and ethanol undergo transesterification) of EPA-EE are described, emphasizing the potential of green substitute technologies. Then, the method of EPA enrichment is thoroughly detailed, the pros and cons of different methods are compared, and current developments in monomer production techniques are addressed. Finally, a summary of current advanced strategies for dealing with the low oxidative stability and low bioavailability of EPA-EE is presented. In conclusion, understanding the entire production process of EPA-EE will enable us to govern each step from a macro perspective and accomplish the best use of EPA-EE in a more cost-effective and environmentally friendly way.

## 1. Introduction

An estimated 17.7 million people died globally from cardiovascular disease in 2017 alone. Indeed, cardiovascular disease has become one of the most serious health burdens of our time [1]. Currently, statins remain relatively common and effective drugs for the treatment of cardiovascular disease, but the use of drugs can have adverse effects, such as liver damage and ischemic events [2]. Surprisingly, in December 2019, the FDA approved the use of eicosapentaenoic acid ethyl ester (trade name: Vascepa), which is up to 97% pure, as an adjunctive (secondary) therapy to reduce the risk of cardiovascular disease in adults with hypertriglyceridemia [3]. Most eicosapentaenoic acid ethyl esters (EPA-EE) are isolated from marine fishes naturally rich in omega-3 polyunsaturated fatty acids. No adverse effects have been observed in the long-term treatment of cardiovascular disease. Many prospective cohort epidemiological studies have shown that omega-3 PUFAs reduce the risk of cardiovascular disease [4].

Furthermore, the REDUCE-IT trial, which used only high-purity EPA formulations rather than a mix of omega-3 PUFAs, demonstrated a 25% decrease in the overall number of cardiovascular events in people with diabetes, existing heart disease, or both [5]. This demonstrates the distinct and extremely effective health effects of eicosapentaenoic acid compared to alternative omega-3 PUFAs. In addition to cardiovascular benefits, eicosapentaenoic acid has various additional promising health benefits (Table 1), such as anti-inflammatory activity, neuroprotective effects, alleviation of depression or major depression, and slowing of diabetes mellitus. Numerous researchers have conducted reviews and attempted to elucidate the mechanisms of health effects of eicosapentaenoic acid. Still, in general, the highly pleiotropic nature of eicosapentaenoic acid may be the answer [6,7].

It is because of the various roles that omega-3 (ω-3) polyunsaturated fatty acids (PUFAs) play in promoting health and reducing disease risk that they have received increasing attention. There are several ω-3 PUFAs, including α-linolenic acid (ALA; 18:3 ω-3), stearidonic acid (SDA; 18:4 ω-3), eicosapentaenoic acid (EPA; 20:5 ω-3), docosapentaenoic acid (DPA; 22:5 ω-3), and docosahexaenoic acid (DHA; 22:6 ω-3). Among them, the most widely studied are EPA and DHA, which are commonly studied as a whole for processing and functionality [8,9]. These PUFAs are commonly present in the livers of lean white fish, the blubber of marine mammals, and the body fat of poly fatty fish. Up to 4% of the vegetable oils we consume are converted into LC-PUFAs in the body, making it necessary to take EPA and DHA supplements daily.

With the better therapeutic and preventive effects of high-purity EPA-EE for specific diseases, there is an increasing interest in the production process of EPA-EE, how to achieve high purity, and how to make it optimally bioavailable [10,11]. Nevertheless, the manufacturing method and the degree of processing determine the EPA and DHA content of fish oil preparation, the form of the eicosapentaenoic acid present affects its bioavailability, and the addition of additional ingredients such as antioxidants affects its subsequent quality [12]. Therefore, to achieve optimal EPA-EE usage, scientists must carefully manage the choice of raw materials, processing, and subsequent encapsulation and utilization to provide consumers a fish oil preparation with a healthier and more excellent eicosapentaenoic acid content.

Until now, limited attention has been paid to EPA-EE; numerous studies have approached it with DHA as a highly concentrated omega-3 PUFA mixture. In this review, we present the first detailed description of the full production process of EPA-EE. First, in previous studies, we analyzed the recent progress of various production processes, such as fish oil extraction, refining, and ethyl esterification. Particular attention was paid to which green methods could be implemented instead of conventional chemical processes. The review also noted the effects of different processes on EPA content, the degree of oxidation, etc.

Furthermore, herein, we describe in detail how the EPA-EE mixture obtained from the abovementioned processing can be concentrated to obtain higher concentrations of EPA-EE. The advantages and disadvantages of different enrichment methods are compared, and the combinatorial application advantages of different enrichment methods are described. Here, the disparities between the physicochemical properties of EPA and DHA are compared to explain the difficulties and challenges of isolating EPA-EE monomers. Recent techniques applied to the preparation EPA-EE monomers, such as simulated moving bed chromatography and countercurrent chromatography, are also described. Finally, we review the recent research on improving the bioavailability and oxidative stability of EPA-EE due to the low oxidative stability and relatively low bioavailability of fish oil ethyl esters rich in long-chain polyunsaturated fatty acids and outline the different storage forms of high-purity ethyl ester fish oils. To the best of our knowledge, this review provides a more direct and extensive account of the fish oil ethyl ester production process. We describe how EPA-EE is transformed from a raw material to an exceptionally concentrated health food or pharmaceutical product.

**Table 1 molecules-28-00672-t001:** Health Benefits of eicosapentaenoic acid.

Health Effect	Reference
Reduction in depression or major depressive disorder	[13,14]
Potential beneficial effects on atherosclerotic plaques	[15]
Neuroprotective effects after stroke	[16]
Control inflammation and tissue homeostasis	[17]
Cardiovascular benefits	[4]
Improve diabetes	[18]
Improve dry eye syndrome	[19]
Benefits for the prevention and management of skin diseases	[20]

## 2. EPA-EE Formation Progress

### 2.1. Fish Oil Extraction

#### 2.1.1. Traditional Fish Oil Extraction

Fish lipids are triacylglycerols, which are rich in nutritional value due to long-chain fatty acids, among which marine fish contain more EPA and DHA than freshwater fish. The contents and profiles of fatty acids in fish species from different regions around the world over 10 years are shown in Table 2. According to results from the last 10 years, Brazil’s marine fish had high levels of both EPA and DHA [21]. Saturated fatty acids were abundant in fish from Chinese offshore basins such as the Pearl River Estuary [22] and the Yangtze Basin [23]. Large quantities of PUFAs have been found in Czech Republic fish [24]. All these results indicate that both fish variety and breeding environments, such as seasons, living conditions, habitats, and age, can influence the contents and profiles of fatty acids in fish oil [25,26]. Fish and fish byproducts were utilized as raw materials to extract fish oil in the 19th century. The raw material was boiled, followed by pressing or crushing with rocks, then filtered or centrifuged to obtain crude fish oil. An oil–water emulsion was easily formed during the traditional fish oil extraction process, and the emulsion was too stable to be separated by filtering or centrifuging [27]. Additionally, long-chain polyunsaturated fatty acids with double bonds were reported to be unstable under the high temperature, so EPA and DHA were vulnerably oxidized to form some hazardous peroxides, such as aldehydes, ketones, acids, esters, alcohols, and short-chain hydrocarbons [28].

The organic solvent extraction method is another conventional technique. Organic solvent extraction typically requires a procedure for eliminating water as a first step to prevent the difficulty in separating emulsions caused by failure to remove water by compressing, cooking, and crushing [10]. Grinding is typically required simultaneously to enhance the sample’s surface area in contact with the organic solvent. Fish oil is hydrophobic; thus, organic solvent extraction is preferred and efficient. Although this method requires a lengthy pretreatment period, it has the potential to be expanded to the plant scale and produces a sizable yield [29]. The disadvantages of solvent extraction have not yet been eliminated despite the development of Soxhlet extraction and accelerated solvent extraction (ASE), which can be used to cut down on time and solvent use [30]. Organic solvent residue is the main issue associated with the use of fish oil. Owing to their toxicity, which is undesirable in terms of ecology and environmental sustainability, organic solvents are no longer widely used in the food industry. Now, they are employed in laboratories on a smaller scale more frequently, typically for analytical or comparative purposes [12].

**Table 2 molecules-28-00672-t002:** Comparison of fatty acid profiles in fish species from different regions around the world over 10 years.

Year	Location	SFAs	MUFAs	PUFAs	EPA	DHA	Reference
2020	Pearl River Estuary, China	38.90	31.10	20.90	5.72	7.44	[22]
2019	Chile	40.82	39.44	18.71	1.49	2.67	[31]
2018	Central Europe	24.76	39.99	34.14	3.74	8.73	[32]
2018	Czech Republic	23.69	37.50	37.28	4.43	13.15	[24]
2014	Brazil	25.40	31.10	31.51	7.85	19.20	[21]
2014	Panama	39.80	31.80	28.30	6.24	15.30	[33]
2014	Yangtze Basin, China	33.05	34.47	32.81	2.53	7.49	[23]
2013	Egypt	25.60	48.30	26.10	1.15	10.30	[34]
2012	Southern Italy	43.60	24.90	31.50	6.82	13.80	[35]
2012	India	37.10	23.70	32.30	4.28	3.08	[36]
2012	Black Sea	31.30	28.40	27.00	6.30	14.50	[37]

#### 2.1.2. Emerging Green Fish Oil Extraction Technology

Researchers are currently focused on the environmentally friendly methods often used to extract vegetable oils to achieve the goal of environmental protection [38]. Table 3 lists the fundamentals, benefits, and drawbacks of green extraction methods such as microwave-assisted extraction (MAE), ultrasound-assisted extraction (UAE), enzymatic methods, and supercritical fluid extraction (SFE).

MAE is an extraction technique that combines the use of microwave energy and conventional solvent extraction. Compared to conventional procedures, MAE results in the rupture of fish tissue, allowing the oil to be liberated and transported to the solvent more quickly and effectively. Compared with the organic solvent mixture of chloroform and methanol (2:1, *v*/*v*) reported by Folch et al. [53], the advantages of MAE are as follow: lipids can be extracted from up to 12 samples at once in up to 90% less time with less residue and with up to 25% less solvent consumption relative to the mass used. Additionally, environmental toxicity is reduced by using ethyl acetate in place of chloroform [54]. The fish oil raw material impacts the effectiveness of MAE’s extraction process, and the choice of the system’s technical parameters also significantly impacts the production of fish oil from MAE. By adjusting parameters including irradiation time, microwave temperature, and microwave power, other researchers further optimized the response of surface methods, and fish oils rich in Omega-3 PUFAs have been successfully extracted from eel [41] and salmon [42], with EPA accounting for 11.11% and 9%, respectively. The EPA content determined by the Soxhlet method of MAE was low in all fish species except for sockeye, probably due to heat exposure and the low lipid yield of wet fish tissues [40]. Consequently, the first and most significant challenge to be solved by this technique is the oxidative loss of EPA caused by temperature exposure.

UAE is widely used to extract bioactive substances from cash crops such as fruits and vegetables [43,44]. As the importance of seafood is increasing, researchers are also concentrating on using UAE to extract contaminants and natural products from fish and aquatic organisms, among others [45]. UAE breaks the cell walls using sonic cavitation, which increases the solvent’s permeability and makes it simpler for the oil to be released. Naturally, UAE still has the benefit of quicker extraction times and less solvent usage compared to conventional techniques. The primary benefit of UAE over MAE is that it can be carried out at moderate temperatures, minimizing the loss of bioactive compounds and minimizing damage to the extract from high temperatures [38]. The livers of Cobia make excellent starting materials for the production of EPA/DHA-enriched oils; Kuo et al. applied homogenizer extraction plus sonication to obtain EPA and DHA contents of nearly 25%. The method is quite efficient [46]. UAE is also applied to raw materials such as rainbow trout [55], tuna [56], and salmon [57]. Although MAE and UAE are hardly ever used in fish oil extraction, owing to their advantages over conventional techniques, they are expected to be increasingly applied in the future.

The enzyme method is the fish oil extraction method that best fits green chemistry practices. It uses exogenous protein hydrolases (such as proteases, exopeptidases, or endopeptidases) to extract oil from fish feedstock without needing organic solvents or elevated temperatures. As Baltic herring is a small class of fish and difficult to process, Aitta et al. attempted to extract fish oil by enzymatic methods. They chose three commercially available protein hydrolases (Alcalase, Neutrase, and Protamex) to extract fish oil and byproducts with two processing durations (35 and 70 min). Extraction with *Protamex* for 35 min resulted in the best fatty acid composition with the highest content of EPA and DHA, in addition to increasing oxidation [47]. This study, while requiring further enhancement and optimization of the quality of the extracted products, indicates the potential of protein hydrolases for fish oil extraction. Sangkharak et al. used one-step extraction saponification combined with an enzyme concentration method to extract and concentrate polyunsaturated fatty acids from Nile tilapia waste; high concentrations of EPA and DHA were obtained when *P. fluorescens* (3.6%) and *Thermomyces lanuginosa* (5.3%) lipases were applied [48]. Additionally, native lipase from yeast *Cryptococcus* sp., was used to extract sardine oil for the first time, with a hydrolysis ratio of 83.7% obtained after 72 h [49]. Adding exogenous enzymes makes the digestion process more controllable and reproducible but does not easily handle scaling problems or upgrade the reuse of enzymes. Overall, enzymatic hydrolysis is an ideal way to recover oil and protein from fish and fishery processing waste.

The scale of extraction is small and expensive with enzymatic technologies, notwithstanding their environmental friendliness. Although the residues and quantities are minimal, prior MAE and UAE still necessitate the use of organic solvents. As a green technology that does not use organic solvents and is suitable for large-scale production, SFE appears to be the most promising overall technique. The common supercritical fluid is carbon dioxide, which exhibits excellent solvent capacity and transport performance. This technology has been summarized by many researchers in reviews of fish oil extraction [12,27,30,38,58]. Recently, after the temperature and pressure of SFE were investigated with fish oil extraction from salmon trimmings, the final content of PUFA extracts was increased to up to 26.1% [50]. In addition, subcritical dimethyl ether extraction (SDEE), which is comparable to supercritical fluid extraction, extracts oil from high-moisture tuna liver without freeze drying at relatively low pressure, with compositions of EPA and DHA in the extracts of close to 28% [51]. When extraction was conducted on Myctophidae fish, fatty acid profiles showed that the level of omega-3 PUFAs under SFE treatment was significantly higher than under wet pressing (*p* < 0.05) [52]. Briefly, supercritical fluid extraction has a wide range of applicability despite the drawbacks of expensive equipment and high energy consumption.

In addition to the abovementioned methods, some novel approaches have emerged, such the use of novel solvents. Ciriminna et al. used limonene, a green biosolvent obtained from waste orange peel, to extract anchovy oil at ambient temperature and pressure [59]. This system allows for recycling, eliminates unpleasant odors in the fishery, and contributes to worker health [60]. According to laboratory research, green extraction techniques significantly replace conventional ones. However, these methods require further research. Techniques for preprocessing and the extraction method itself must be improved.

### 2.2. Fish Oil Refining

Once extracted, fish oil is purified because it contains insoluble impurities, phospholipids, free fatty acids, water, primary oxidation products, minerals, pigments, and even persistent organic contaminants, which we call crude oil [10]. Therefore, to achieve acceptable quality requirements for human consumption and meet the standards for EPA/DHA-rich fish oils, fish oil refining needs to address two main difficulties: (1) removal of impurities from the oils and fats and (2) stabilization of fish oils to prevent their oxidation and off flavors.

The procedure is usually as follows: degumming, neutralization, bleaching, and deodorization. The typical refining process starts with degumming using 1% phosphoric acid, which is neutralized with 1M NaOH, followed by centrifugation-based deacidification, washing and drying with hot water, bleaching with a combination of adsorbents, and finally, steam distillation under vacuum to remove odors [61]. Many studies have generally followed the traditional refinement process; however, researchers have made new attempts for each part.

Degumming, the initial step in extracting fish oil, removes phospholipids and some trace amounts of metals and mucilage. Fish oil degassing is normally achieved by either chemical or enzymatic degassing [61,62,63]. Phosphoric acid was reported to be of higher degumming quality for crude sardines, and no significant difference (*p* < 0.05) was found in the PUFA compositions after degumming [64]. Enzyme degumming is a green alternative; Lamas et al. used Lecitase Ultra^®^ phospholipase A_1_ and LysoMax^®^ oil acyltransferase to degum different fish oil samples; there was no significant difference (*p* < 0.05) in the fractions of polyunsaturated fatty acids when using enzymatic degumming, and enzyme degumming resulted in better color levels than conventional degumming [62]. Compared with fish oil degumming, the plant oil degumming process combines chemical degumming and enzymatic degumming and uses a chelating agent, disodium EDTA with emulsifier, for soft degumming; membrane technology, which can be used in fish oil degumming, has also been investigated in degumming process research [65].

Neutralization, also known as deacidification, eliminates free FA and residual phospholipids by reacting with a base (usually soda) to reduce the acidity the oil. Fish oils contain high levels of free fatty acids, so deacidification is critical in fish oil refining [65]. Along with traditional alkali, solvent extraction and membrane-assisted pre-extraction combined with methanol extraction can also be used to solve the problem of deacidification of degummed fish oil [66]. The deacidification process is environmentally friendly, using enzymes and natural solvents. Utilizing the commercial Novozym 435, high-acidity crude fish oil can be enzyme-digested to reduce acid values while increasing EPA and DHA concentration [67]. In addition, the majority of natural antioxidants in oils can be lost during the deacidification process, mainly because of high temperature, while betaine monohydrate-based natural deep eutectic solvents have extensive potential for deacidification of oils. Zahrina reported that free palmitic acid was efficiently extracted with betaine monohydrate-based natural deep eutectic solvents, and most of the natural antioxidants could be preserved in refined palm oil. The possible reason was that the distribution coefficients of free palmitic acid were improved, while those of natural antioxidants were significantly reduced in betaine monohydrate-based natural deep eutectic solvents [68]. More importantly, Sander et al. showed that DES can be applied to vegetable and animal oils and proposed scaling-up criteria for waste oil deacidification [69].

Utilizing activated clay or carbon to absorb pollutants such as pigments, trace metals, oxidation products, and other materials is known as bleaching. Treatment with solid sorbents is a method frequently employed to bleach fish oil. To improve color and reduce oxidation levels in crude fish oil, the sorbent, and activated carbon, should be used under ideal conditions [70]. When combined with vacuum conditions for decolorization, adding 10.18% adsorbent—which contains 70% activated white clay—can produce low oxidation values, with 44.4% decolorization and up to 94.12% heavy metal removal [71]. The fatty acid composition is largely unaffected by the decolorization process.

Moreover, Igansi et al. investigated isotherms, as well as kinetic and thermodynamic profiles, for the adsorption of carotenoids and peroxides from catfish oil extracted from head waste using a mixture of activated clay and activated carbon as adsorbents, providing a mathematical model that can be used as a reference for fish oil decolorization [72]. It should also be noted that for treatment of persistent organic pollutants, the efficacy varies by material. Studies have shown that carbon-based adsorbents can help reduce 99%, 70%, and 27% of polychlorinated dibenzo-p-dioxins, hexachlorobenzene, and polychlorinated biphenyls, respectively, while silica-based adsorbents are not as effective as the former [73]. In summary, conventional bleaching processes using solid adsorbates are cost-effective in industrial applications. Therefore, during this stage, additional attention was given to the optimization process of the amount of added adsorbent and the adsorption temperature necessary to maximize the removal of unwanted compounds while maintaining a high PUFA content.

Deodorization is the removal of odorous compounds. At the end of this step, the final product is a light-colored, odorless, high-quality refined oil. Steam distillation is the most often employed process to remove unwanted odor components, mainly aldehydes, ketones (lipid oxidation products), and residual FFA [65]. After traditional chemical refinement of the tuna byproducts obtained by enzymatic hydrolysis, Oliveira et al. concluded that the optimal deodorization conditions for PUFA-rich tuna byproduct oils are 160 °C for 1 h and 200 °C for 1 h (steam temperature and time), which can better preserve the content of EPA and DHA [74]. Researchers have worked to find gentler, lower-energy options as an alternative to high-temperature steam deodorization. Fang et al. provided an option for deodorization via a nanofiltration membrane process. The nanofiltration membrane removed most of the volatile fractions, with a removal rate of 80%. The odor activity values of the volatile components of tuna oil and squid oil were significantly reduced from 125.23 to 17.47 and from 129.76 to 10.06, respectively (*p* < 0.05). On the other hand, as predicted, EPA and DHA content increased [75]. Song et al. additionally attempted to replace the industrially used molecular distillation and vapor distillation with liquid–liquid extraction, green tea polyphenol processing, and solid-phase adsorption of activated clay, zeolite, or diatomaceous earth. Comparing these techniques in terms of physicochemical properties, sensory properties, saturation, and cis/trans conformation of fatty acids and volatile compounds, the results show significant differences (*p* < 0.05). More importantly, results also indicate a positive effect of purification on fish oil polyunsaturated fatty acids [76].

Researchers examined how the four-stage refining process affected changes in the fatty acid composition, volatile compounds, and some fundamental physicochemical indicators of fish oil (such as peroxides and p-anisidine). They concluded that refining increased the amount of PUFAs and the quality parameters of fish oil [61,77,78]. It is critical to assess the impact on the quality of fish oil during the various refining processes to better meet the requirements for the next step of enrichment and concentration and possibly to prevent the oxidation of fish oil, thus maximizing the value of EPA/DHA-rich fish oil.

### 2.3. Production Process of EPA-Rich Ethyl Ester Fish Oil

Since the recognition of the distinct functional properties of EPA in omega-3 PUFAs, fish oils with a single component of EPA have become increasingly importance. Eicosapentaenoic acid ethyl ester (EPA-EE) can be obtained by reacting refined natural fish oil with ethanol under certain conditions. Ethanolysis can also accomplish EPA-EE concentration.

A complicated mixture of triglycerides (TAGs) used as feedstock for ethyl esterification primarily contains PUFAs attached to the sn-2 position of the glycerol backbone. It is necessary to dissolve the glycerol backbone of TAGs to release the free EPA before the oil can be enhanced with the most valuable EPAs. Under alkaline or acidic conditions, the TAG backbone is easily broken by esterification with methanol or ethanol (alcoholics) to produce methyl or ethyl ester derivatives of EPA (EPA-MEs or EPA-EEs) [79]. EPA-MEs are commonly used for analytical purposes, and in the food industry, ethanol is preferable, as it is certified as safe by the FDA [12]. These are often used on an industrial scale to prepare ethyl ester fish oils with good economics. However, chemical transesterification can harm the oxidative stability of the oil [80]. Poisonous chemical solvents can also cause damage to the environment. To overcome these problems, Piao et al. used Brønsted acidic ionic liquids (BILs) as a green alternative solvent to catalyze the transesterification reaction of fish oil and ethanol with ultrasound assistance, resulting more esterified EPA in this environment [81].

Lipases are unique enzymatic tools that can selectively improve the natural composition of lipids and provide specific bioactivities while drawing widespread attention and interest from the food and pharmaceutical industries for their high efficacy, greenness, and low energy consumption [82,83]. In contrast, 35 lipase types in the study were found to have a preference for EPA triacylglycerol; and lipase from alkali hyperthermophilic *Archaeoglobus fulgidus* (2ZYR) showed the strongest binding preference for EPA triacylglycerol. This undoubtedly provides extra impetus to realize EPA-EE concentrates [84]. Another study revealed that the liquid lipase CAL-A was more capable of enriching EPA (about 40%) at ethanol/oil ratios higher than 8:1. It was also indicated that lipase selectivity was linked to the structure of the enzyme and its interaction with the substrate and solvent [85]. Therefore, CAL-A can be considered an ideal lipase for concentrating EPA via ethanol [86].

Additionally, liquid enzymes and immobilized lipase have advantages in the ethanolysis of fish oil [87]. Although Guo et al. used free PUFAs as raw material, the immobilized lipase Novozym 435 achieved up to 94% ethyl esterification conversion [88]. When immobilizing lipase on an inert support in an unusual reaction medium, Borges et al. found that ethanolysis of sardine oil with Duolite A568-*Thermomyces lanuginosa* derivative in cyclohexane produced 24% EPA-EE and 1.2% DHA-EE, which displayed a high selectivity index of 20 times that of DHA-EE [89]. Moreover, choosing a suitable immobilizing material can lead to higher thermal stability and a preference for enzymes. Mohammadi et al. found that *Rhizomucor miehei* lipase and its derivative pairs immobilized by different methods could differentiate between EPA and DHA and favored EPA [90]. With the diversification of immobilizing carriers, researchers have discovered that the key to producing EPA-EE depends on the carrier used for immobilization, which is the determining factor for activity. Cipolatti et al. synthesized carriers to immobilize *T. thermophila* lipase (TLL) using microemulsion technology. TLL immobilized on PEGylated polyurethane particles was efficiently used for ethyl ester production from fish oil compared to free TLL; the highest selectivity value of 45.8 (EPA/DHA) was obtained for the PU-PEG4000-PEI20 derivative [91].

Enzymatic synthesis of fish oil in the ethyl esterification process is an emerging field in which researchers are working to discover additional specific lipases or develop improved immobilization methods and materials for efficient enzyme utilization. Because the ethyl esterification of fish oil of fish oil serves both as a step in the formation of EPA-EE and as a tool for concentration, these two objectives are inextricably linked. In conclusion, greater thought should be given to green technology in forming EPA-EE to make the process more environmentally friendly and long-lasting. Figure 1 summarizes the main processing steps and the latest technology in the EPA-EE formation process.

## 3. EPA-EE Mixture Enrichment and Its Monomer Preparation

### 3.1. EPA-EE and DHA-EE Physical and Chemical Properties

To obtain high-purity EPA-EE monomers, it is necessary first to obtain a mixture rich in EPA and then separate DHA-EE, which has various physicochemical properties extremely similar to those of EPA. The difference in properties between the two is the critical point for separating high-purity monomers.

Omega-3 fatty acids, called n-3 fatty acids or ω-3 fatty acids (n-3 FAs), are a heterogeneous group of fatty acids with a double bond between the third and fourth carbon atoms, starting from the methyl end (from the ω-1 carbon atom). Some researchers mistakenly believe that long-chain (LC)n-3 PUFAs and omega-3 fatty acids have the same meaning, which is wrong because “omega-3 fatty acids” is a general term [92]. EPA and DHA, the most popular omega-3 polyunsaturated fatty acids, both have double bonds starting from the third carbons at the methyl terminal, and the carbon–carbon double bonds in the molecule are in a non-conjugate system, both in the cis configuration. On the other hand, the ethyl esters of EPA and DHA have an ethyl group attached to the carboxyl end. These products generally appear as a colorless or pale yellow clarified oily liquid with a fishy odor and are insoluble in water and readily soluble in organic solvents such as ethanol, ether, chloroform, etc. Because they contain multiple double bonds, they are prone to oxidation and degradation in air. The only structural difference is that EPA-EE has one fewer double bond and two fewer carbons than DHA-EE (Figure 2).

Additionally, the properties shown in the table above (Table 4) are used to calculate the physical and thermodynamic properties of pure compounds [93]. Rossi et al. used the group contribution fragmentation method applied to organic compounds to calculate the critical properties. In contrast, the Lydersen–Forman–Thode method was applied to calculate the normal boiling point [94]. The critical temperature, the critical vapor pressure, and the normal boiling point are close. The critical temperature was 562.84 °C and 592.84 °C, and the normal boiling point was 407.51 °C and 434.11 °for C EPA-EE and DHA-EE, respectively. Moreover, the critical vapor pressure of EPA-EE was only 0.91 (105 Pa) higher than that of DHA-EE.

Therefore, the similarity of the properties of EPA-EE and DHA-EE poses a great challenge to researchers who wish to obtain monomers, and this challenge is expected to continue. Figure 3 summarizes the advantages of each EPA-EE enrichment technique and briefly shows the results of the combination of individual techniques.

### 3.2. EPA-EE Mixture Enrichment

#### 3.2.1. Urea Complexation

AS the procedure enables the processing of large quantities of material with basic equipment using cheap solvents and mild conditions, in addition to its low cost, urea complexation (UC) appears to be one of the most suitable methods for ω-3 PUFAs enrichment on an industrial scale. In the process of urea inclusion, when fatty acids and urea are fully mixed and crystallized at a certain temperature, saturated and monounsaturated fatty acids first combine with urea to form a hexagonal crystal inclusion complex. At the same time, EPA and DHA are retained in the solution [11]. EPA and DHA are then obtained by filtration and nitrogen blowing. Following the basic features of urea, this approach only applies to the enrichment of free ethyl esters in fish oil.

Along with inclusion formation, different organic solvents are commonly used to improve the contact between urea and fatty acids. Ethanol and methanol are the most widely used solvents for this purpose [49]. However, due to the microtoxicity of methanol, ethyl fish oil production is preferred in the study of ethanol as a solvent. Although the UC method is an old-fashioned technical tool, it still has tremendous applications in recent years. Applying the response surface methodology (RSM) to optimize the independent variables of the UC process, Dovale-Rosabal et al. obtained high levels of EPA and DHA in refined commercial salmon oil concentrated under optimized conditions (6.0, urea: FA content ratio; −18.0 °C, crystallization temperature; 14.80 h, crystallization time; 500 rpm, stirring speed). The total omega-3 content increased from 13.78% to 80.51%, and EPA content increased by 4.1 times (from 7.53% to 31.2%) [95]. Similarly, Pando et al. employed RMS to analyze the effects of the ratio of urea to fatty acid, crystallization temperature, crystallization time, and stirring speed on EPA and DHA enrichment to obtain higher EPA and DHA concentrations from rainbow trout processing byproducts; ultimately, 71.52% EPA and DHA was achieved under optimal conditions (EPA accounted for 20.50%) [96]. On the other hand, in addition to employing RSM, it may be helpful to apply the gradient cooling UC method to achieve a higher purity of PUFAs [97].

Most studies have focused on fish oils with a fixed fatty acid composition without considering the regularity of the variation of different fatty acids during the process. Zhang et al. concluded that the ratio of urea to fatty acids affects the recovery of fatty acids obtained by the UC method. This recovery decreases with an increasing ratio. Under optimal conditions (urea/FA-EE mass ratio of 2.38:1 at 15 °C for 2.5 h of crystallization), the concentrations and recoveries of PUFAs (EPA, DPA, and DHA) were 71.35% and 82.31%, respectively, and the models for EPA, DPA, and DHA were highly significant (*p* < 0.01) at the 99% confidence level (EPA purity of 28.97%) [98]. However, the study also suggested that EPA was more easily entrapped in urea than DPA and DHA, which implies that using the UC method to obtain high concentrations of EPA-EE requires more thought. An alternate type of oil could possibly reduce the interference of other PUFAs with EPA-EE enrichment, resulting in a higher concentration. Gonzalez-Fernandez et al. selected five types of oil, including commercially available tuna oil. After optimizing the urea-to-fatty acids ratio, a DHA algal oil concentration of 98% was obtained.

In contrast, the other EPA algal oil and DHA fish oil were not as prominent, indicating that it is critical to choose a suitable oil when concentrating PUFAs through UC methods [99]. Thus, high-purity PUFA enrichment can be achieved by selecting the right oil for enrichment, particularly the right fatty acid composition and ratio. For example, EPA-rich algal oil or specific microbial culture oils may be better choices for purification [100,101].

Despite the benefits of the UC technique, it would not be ideal for the food or pharmaceutical industries due to the reported creation of two carcinogens, i.e., ethyl and methyl carbamate. However, as the study progressed, Vazquez et al. gradually grasped the regularity of ethyl carbamate (EC) generation; EC increases as the temperature increases, whereas EC was effectively removed from the product using an aqueous washing step, and the optimal conditions for the enrichment of PUFAs was 21 °C with 3 g urea and 0.4 mL ethanol [102]. After that, the authors developed a different method for PUFA enrichment by UC employing a green food-grade solvent, which could effectively replace ethanol and hexane with propionic acid and α-pinene, and no EC generation was detected even before aqueous washing, which undoubtedly offered a fresh solution to the problems arising in association with the UC method [103]. There are various strategies available to remove carcinogens formed by urea, the simplest being urease [104]. In summary, there is still considerable potential for the UC method to concentrate EPA-EE. However, the question remains as to how to easily and economically recycle used urea in an environmentally friendly manner.

#### 3.2.2. Low-Temperature Crystallization

Low-temperature crystallization is similar to the urea inclusion method, and both have been applied to concentrated ethyl esters and free fish oil. This methodology enables the enrichment of PUFAs through cooling to remove high-melting-point compounds (i.e., saturated fatty acids). The melting point of fatty acids of a given chain length decreases with increasing unsaturation. Therefore, at low temperatures, saturated fatty acids crystallize into a solid phase, whereas PUFAs are retained in the liquid phase [11]. The use of solvents during low-temperature crystallization promotes crystal formation and increases the yield and purity of the crystals.

The factors affecting the cooling crystallization method are the oil composition, the crystallization temperature, the mobility of the molecular species in the oil (influenced by the polarity of the solvent), the ratio of oil to solvent, and the rate of cooling. The lower the temperature, the higher the concentration of PUFAs in the liquid fraction [27]. However, depending on the balance between fixed costs and the purpose of PUFA enrichment, the choice of oil and solvent is particularly critical. Utilizing solely hexane for enrichment yields a better purity than utilizing various hexane and acetone combination ratios [105]. The highest recovery of EPA and DHA was achieved when just acetone was utilized as the organic solvent for the concentration of Cobia liver oil. The resulting concentrate contained 71.23% EPA and DHA (16.81% EPA) [106]. However, when using only acetonitrile, the tuna oil-to-solvent ratio, crystallization temperature, and time were all improved to produce a purity of 79.6% with a much lower solvent ratio (1:10) [107]. Moreover, the use of methanol as a solvent and the addition of α-tocopherol is a useful way to increase the concentration of PUFAs in order to prevent oxidation while cooling the crystallization to enrich PUFAs [108].

In addition to the effect of the liquid phase composition on the concentration of PUFAs in each factor, it is advantageous to improve the low-temperature crystallization method to consider how the process of phases evolves. Mass transfer theory was used to determine the effects of temperature and time on the liquid phase mass produced by crystallization at low temperatures. This model was then used to determine the composition of EPA and DHA in the liquid phase. According to the research results reported by Morales-Medina et al., crystallization at −85 °C for 24 h produced a PUFA concentrate of up to 80%, of which 38.7% was EPA [109]. Research has also been conducted based on mathematical models of diffusion–reaction theory to predict the mass evolution in the cooling crystallization process [110].

In conclusion, although low-temperature crystallization enrichment has the advantages of a simple process, convenient operation, and low cost, the application of the method is still limited, as it has considerable required for low-temperature cooling equipment, extensive use of organic solvents, and the possibility of residuals.

#### 3.2.3. Molecular Distillation

Molecular distillation (MD) is a method of thin film evaporation operating at high temperatures and high vacuum levels, whereby reduced ambient pressure allows for the successful physical separation of thermally unstable compounds at atmospheric pressure [111]. In the extraction of polyunsaturated fatty acids from fish oil by molecular distillation, different fatty acids have different boiling points under specific vacuum conditions depending on their carbon chain length and saturation. Molecular distillation is performed in multiple stages using controlled MD temperature, pressure, and flow rate adjustments. Products with different EPA and DHA ratios are available to meet the usage requirements of different targets.

To optimize the operation of molecular distillation, researchers often use RSM and artificial neural network approaches. Some researchers have developed specific simulation software models to simulate the complex dynamics of the MD process [112]. Limited studies have been performed on the purification of EPA-EE using molecular fractionation alone. Using molecular distillation, Rossi et al. optimized the separation process of ω-3 fatty acid ethyl esters obtained from squid oil. The separation proceeded in two stages, with the first stage operating in the temperature range of 100 to 120 °C and the second stage in the temperature range of 120 to 140 °C. A mathematical model was developed based on the phenomenon of mass transfer occurring in an isothermal separation of components. The Langmuir–Knudsen instanton equation is applied to express evaporation dynamics. The mathematical model was solved in the investigated working area and numerically validated using experimental data [94]. Later, researchers continued to develop a prediction model by representing the concentration process by molecular distillation of ω-3 compounds and utilizing artificial neural network techniques to forecast the behavior of the molecular distillation process in two stages (e.g., ester ethyl of EPA and DHA obtained from squid oil) [113].

Typically, the enrichment of PUFAs in fish oil is more commonly selected using MD and other purification methods. In this way, MD can remove monounsaturated fatty acids that are difficult to remove by other methods [98]. The advantages of molecular distillation are clear. It can be carried out without using organic solvents and fed continuously at different scales [112]. Furthermore, it is suitable for high boiling points and thermally unstable substances. However, the upfront expenses associated with sophisticated instrumentation, as well as the ongoing costs of working at high temperatures and high vacuum, may restrict the use of MD.

#### 3.2.4. Enzymatic Purification

As we mentioned, enzyme technology has been widely applied in the specialized food and pharmaceutical fields, and various enzymes have been developed for industrial purposes. In the various processes for the efficient extraction of omega-3 PUFA concentrations from fish oil, both in the extraction and refining of fish oil, enzymatic methods provide a strong impetus as a safe and efficient means to advance the enrichment of high-purity EPA-EE.

Many lipases are more selective for EPA during ethyl esterification of fish oil, such as *Thermomyces lanuginosus* lipase [114]. Gao et al. identified an organic solvent-resistant OUC-lipase 6; an alkaline enzyme that has good regioselectivity for improved enrichment of EPA [115]. It has been observed that *T. thermophila* (TLL) and *Rhizomucor miehei* (RML) are more EPA-selective than *Candida antarctica B* (CALB). During the selective ethanolysis of fish oil using immobilized lipases, Moreno-Perez et al. found that the key to the enrichment of EPA-EE but not DHA-EE depends not only on the type of lipase but also on the immobilized carrier, the physicochemical modification of the immobilized lipase derivatives, and the solvent used. Although all three lipases, i.e., CALB, TLL, and RML, were immobilized by anion exchange and hydrophobic adsorption, EPA-EE was released 20-fold faster than DHA-EE when hydrophobically attached to Sepabeads C18 by TLL in cyclohexane. After further modification of TLL by polyethyleneimine polymer, 80% of EPA was released as ethyl ester after 3 h of reaction at 25 °C at an initial selectivity 20 times higher than that of DHA. At this point, a highly enriched EPA-EE (>95%) was obtained, with DHA-EE accounting for less than 5% [116]. When lipase was adsorbed on the same hydrophobic carrier, Sepabeads C18, as in the above study through an open active center, Moreno-Perez et al. showed that *Ultra lecithin* (a phospholipase with lipolytic activity) immobilized by this method exhibited extraordinary EPA selectivity. The ethanolysis of EPA catalyzed by this immobilized enzyme was 43 times faster than that of DHA. In contrast to previous studies, in this way, the best selectivity can be obtained in a solvent-free system instead of using organic solvents, and it is possible to obtain almost 100% pure EPA-EE in the first stage of sardine oil hydrolysis [116]. Attempts to synthesize EPA-EE in solvent-free systems were continued by Aguilera-Oviedo et al. They used the predictive simulation software Conductor to select the potential solubility of the green solvent for EPA-EE extraction before proceeding with a one- or two-step reaction of ethyl ester, then employed two resting cells obtained in the laboratory (*Aspergillus flavus* and *Rhizopus oryzae*) and the commercial immobilized enzyme Novozym 435 as biocatalysts in the solvent-free system. Results were reported for the three catalysts with yields of 63%, 61%, and 46% in one-step extraction. For the hydrolysis step in the first two-step reaction, Novozym 435, *Rhizopus oryzae,* and *Aspergillus flavus* free fatty acids were obtained at 93%, 83%, and 88%, respectively. In comparison, Novozym 435 showed the highest yield in the esterification step (85%), followed by *Rhizopus oryzae* (65%) and *Aspergillus flavus* (41%). Therefore, it is evident that filamentous fungal resting cells can also be a powerful tool for the esterification enrichment of omega-3 PUFAs, with a lower cost than commercial immobilized enzymes [117].

Some modern techniques can assist in enhancing the extraction reaction rate. Continuous production of DHA and EPA ethyl ester can be achieved with the help of a newly developed ultrasound-filled bed bioreactor using the immobilized lipase Novozym 435 as a biocatalyst. Results show that ultrasonic-filled bed bioreactors have high external mass transfer coefficients and low external substrate concentrations at the surface of the immobilized enzyme. The superior specificity constant of a sonication bath over a normal oscillation bath has also been demonstrated by kinetic modeling, indicating that sonication treatment increases the affinity between the enzyme and the substrate, thus increasing the reaction rate and maintaining a stable 98% conversion [118]. He et al. also used ultrasound for the enzymatic preparation of fatty acid ethyl esters in deep eutectic solvents and found that ultrasound power and temperature synergistically affected ethyl ester transformation up to 93.61%. In particular, a deep eutectic solvent is a cheap and non-toxic green solvent, providing a new possibility to obtain EPA-EE [119].

Traditionally, the enrichment of ω-3 fatty acids has been carried out at high temperatures, which requires high energy input and may lead to product breakdown. Enzyme extraction and enrichment have been investigated as green alternative technologies. Immobilized enzymes are particularly worth using due to their high stability, reusability, and feasibility of separation from the product [120]. The enzymatic enrichment of EPA-EE with immobilized lipases has also been considered as more sustainable and cleaner, and more research has been performed to design more active and stable immobilized enzymes to concentrate EPA-EE from marine byproducts in large-scale and consecutive processing [118,120]. Therefore, it would also be beneficial for us to produce higher-purity EPA-EE in a gentler and more efficient environment.

#### 3.2.5. Membrane Technology

Compared with traditional fatty acid (FA) concentration methods, membrane separation has the advantages of low energy consumption, a compact system, high separation efficiency, convenient scale-up, large effective surface area per unit volume, and low operating temperature [121]. As the separation of FAs depends, to some extent, on the molecular size of the constituents in the FAs mixture, differences in membrane pore diameters are crucial for achieving high purity and quality of FAs. Suitable membranes can be separated based on the membrane’s pore size or chemical affinity to the percolating component.

The most commonly used membrane method for separating FAs is ultrafiltration. Ultrafiltration is a pressure-driven membrane procedure that separates different molecules from a mixture. Ghasemian et al. studied the concentration of ω-3 PUFAs from lantern fish oil by ultrafiltration membrane. A more selective separation of ω3-PUFA was observed for ETNA01PP between the tested membranes. In addition, a Box-Behnken experimental design was used to evaluate the effects of the main parameters affecting the concentration of ω-3 PUFAs. ANOVA revealed a significant effect of these parameters on the concentration of PUFAs. Under optimal process parameters (temperature of 36.19 °C, pressure of 4.82 bar, and stirring rate of 43.01 rpm), a maximum concentration of 35.1058% of PUFAs was obtained [122]. Denser structures or smaller pore sizes can enhance the size-sieving performance of the prepared membrane, leading to higher selective separations. The authors also evaluated the concentration performance of synthetic polyvinylidene fluoride (PVDF) membranes on lantern fish oil ω-3 PUFAs. The effect on concentration was evaluated at different pressures and temperatures, resulting in the best ω3-PUFA concentration (40.4%) at 5 bar and 30 °C [123]. Owing to its superior thermal and mechanical strength, PVDF is a widely utilized polymeric material in the construction of membranes. However, its separation capability does not match the level of industrial demand.

Nonetheless, it would be interesting to integrate PVDF with inorganic particles due to the inherent properties of organic–inorganic compounds, which could improve the separation performance of PVDF. Researchers fabricated asymmetric nanocomposite membranes using PVDF and nanoporous silica particles that exhibited enhanced antiscaling properties. ω-3 PUFA was concentrated up to 53.9% at 15% silica weight under the conditions of 30 °C, 4 bar, and 100 rpm (with 15.07% EPA) [124]. In addition to using the materials mentioned above, some researchers have attempted to separate EPA-EE and DHA-EE by applying deeply crosslinked epoxy nanofiltration membranes, as the membrane flux through EPA-EE is significantly faster than that through DHA-EE by a factor of 1.4. According to researchers, future studies on these membranes will focus on chemical classes including fatty acid esters and their potential application in pressure-driven processes [125].

It is important to introduce membrane technology into the concentration of fish oil PUFAs, as membranes have shown promising applications in the concentration of fish oil omega-3 PUFAs. Membrane separation can be employed as a standalone method for the separation fatty acids or be combined with the hydrolysis reaction process to achieve the simultaneous preparation and separation of fatty acids [121]. We are certain that the use of membrane technology for the separation, purification, and improved synthesis of fatty acids will have ever-growing potential as membrane technology continues to advance. However, it is worth considering when choosing a membrane material.

#### 3.2.6. Combination and Integration of Enrichment Technologies

Other than the techniques mentioned above to concentrate EPA and DHA, there are also many other techniques such as supercritical CO_2_ fluid enrichment and silver ion complexation. While supercritical CO_2_ fluid technology is mainly used for the extraction and refining of fish oils, the production of omega-3 concentrates using supercritical CO_2_ fluid technology is often combined with enzymatic methods or chromatography. Supercritical CO_2_ presents a distinct commercial advantage because of its green process and high-quality products [80]. Another technique is silver ion complexation, which is based on the ability of silver ions to undergo reversible π-bond complexation reactions with substances containing double bonds to achieve separation. In this case, EPA and DHA in fish oil form complexes with silver ions in the aqueous phase to achieve separation from the oil phase. Shanmugam et al. complexed and concentrated EPA-EE and DHA-EE from 18/12 fish oil ethyl ester in a microfluidic reactor with aqueous silver nitrate solution, and only 36 s of contact at 10 °C achieved similar separation in a stirred reactor with continuous stirring for 15 min, with mass fractions of EPA and DHA of up to 42% and 30%, respectively [126].

Although various processes, including urea precipitation, low-temperature crystallization, molecular distillation, supercritical extraction, chromatography, enzyme enrichment, and solvent extraction have their distinct features, process technologies can be selected for different purposes based on the overview and comparison of the relevant parameters (Table 5) [127]. Liquid chromatography is mainly used with other techniques in the concentration step, such as supercritical fluids and silver ion complexation [128].

Because there are limits to single enrichment methods and the applicability of each method varies in practice, it is a good choice to combine two or more methods to give full play to their respective strengths, resulting in higher concentrations of EPA and DHA. Mu et al. used urea complexation to initially enrich ω-3 docosapentaenoic acid (DPA), which was then further purified by preparative high-power liquid chromatography (pre-HPLC) to obtain DPA concentrations up to 93.12% [129]. However, researchers prefer to combine process techniques such as molecular distillation, enzymatic methods, and urea complexation to obtain higher-purity EPA-EE than costly concentration processes such as liquid phase preparation.

Molecular distillation techniques have been widely used in lipid purification. This technique can be used to separate compounds with different boiling points under vacuum, which lowers the evaporation temperature and minimizes residence time, thus allowing for the separation of thermosensitive compounds with minimal thermal degradation [130]. Researchers paired this method with enzymatic procedures and urea complexation as the first step in purification due to its superiority for thermosensitive compounds. The esterification of glycerol and EPA/DHA-rich ethyl esters was first carried out in a two-step enzymatic reaction using Novozym 435. This reaction was then combined with molecular distillation at a lower temperature (140 °C), which resulted in high-purity n-3 PUFAs based on glycerol-triglyceride (98.75%) [131].

Additionally, the optimal operating parameters for each process can be identified by contrasting the two separate processes of urea complexation and molecular distillation. This comparison allows for large concentrations of ethyl n-PUFAs with chemical index values permitted by the current edible oil regulations. In particular, in the second stage of molecular distillation, the concentration in the fraction was reported to be 82.027%, and the ethyl omega-3 PUFA concentration in the residue was up to 95.106% [132].

Lipase-catalyzed reactions such as esterification and ester exchange reactions are widely used to produce ω-3 PUFA ethyl esters. Lipase-assisted processes can be performed under mild conditions with low processing costs and uncompromised product quality. Moreover, lipase is substrate-specific, allowing for the selective formation of high-quality target products. Therefore, the lipase method, in combination with other technical approaches, can provide better concentration results. When oil and ethanol were first esterified using lyophilized recombinant lipase from Proteus vulgaris K80, 82% conversion was obtained with 20% enzyme addition. Later, to enhance the efficacy of the reaction, lipase was immobilized on hydrophobic beads, and the conversion was increased to 86%. Finally, n-3 PUFA EEs were purified up to 92% (with 20% EPA) in conjunction with urea complexation [133]. In addition to urea complexation, some researchers have used alternating cooling crystallization with enzyme exchange for omega-3 PUFA enrichment. Lei et al. screened to determine the optimal Lipozyme TL IM for transesterification, followed by low-temperature crystallization using acetone as the solvent. In the optimized method, alternating low-temperature crystallization (0.1 g/mL oil/acetone, 24 h, −80 °C, precooled Büchner filtration) and transesterification (Lipozyme TL IM, N_2_ flow, 2.5 h, 40 °C) successfully allowed for an increase in the content of omega-3 PUFAs to 43.20 mol%, with EPA content reaching 20.23 mol% [134].

As fish oil quality is crucial for subsequent use, it is vital to guarantee excellent quality indices in the initial processing stages. Results showed that Ag+ bonded to mercaptopropyl silica gel chromatography (AgMSG-CC) was more efficient and stable for the enrichment of EPA and DHA and had a larger-scale purification capacity compared the physicochemical properties, fatty acid composition, and volatile compounds of fish oils obtained by different enrichment methods [135]. This supports the previous claim that the various enrichment techniques result in various purifying products. It would be ideal if enrichment techniques could be combined in a mathematical model based on their distinctive advantages and used to obtain concentrated EPA-EE products. In addition to purity being the first indicator of these enrichment procedures, it is also necessary to thoroughly investigate the impact of various enrichment methods on the quality of fish oil.

### 3.3. High-Purity EPA-EE Preparation

It is well known that EPA-EE has unique physiological functions such as cardiovascular benefits and antidepression activities. Some studies have shown these functions to be better than mixtures with DHA, so there is a need to prepare high-purity monomers of EPA and DHA to achieve improved nutritional use. However, the separation of the two is currently a challenge because of their very similar structures and small differences in properties. The basic strategy to separate individual EPA and DHA is chromatography.

Liquid chromatography has long been the preferred method for obtaining high-purity monomers. Researchers used thiosilver chromatography material to combine silver ion complexation with liquid chromatography and obtained a single isocratic separation of fish oil ethyl esters in 5–10 min with a purity of over 95% for EPA and DHA ethyl esters [136]. Owing to the high cost of silver materials and the possible hazards of heavy metal residues, researchers often use C18 reversed-phase chromatography columns [137]. The conventional preparation of a liquid phase consumes too much solvent and is only semi-continuous. In recent years, researchers have set out to overcome the drawbacks of traditional preparative liquid phases by experimenting with newer techniques, such as simulated moving bed chromatography for the preparation of EPA-EE and DHA-EE monomers. Simulated moving bed (SMB) is a currently accepted technique for continuous chromatography that eliminates the drawbacks of batch chromatography. The operating principle involves the countercurrent motion of the liquid eluent phase and the solid adsorbed phase, which is simulated by moving the column at regular intervals. Such separation processes allow for continuous operation, increasing productivity and conserving fresh solvent, making the process economically viable. BASF applied the SMB process for separation and purification to obtain EPA-EE and DHA-EE with up to 95% purity. They used 19 rods of 40–60 μm C18 as the stationary phase and methanol–water as the mobile phase [138]. Li et al. calculated the EPA-EE and DHA-EE adsorption equilibrium data and kinetic parameters of eight-and-a-half constructed 10 μm C18 columns. The triangle theory design approach was used to determine the separation area, and the two were then perfectly separated in a pilot-scale SMB device. Productivity increases with increased feed concentration. When employing a 100 g/L feed solution, the adsorption productivity is 13.11 g/L/h, the solvent consumption is 0.46 L/g, and the relative purity of both effluxes is above 99% [139]. To reduce the number of columns used and to improve the efficiency, Wang et al. first used a mixture of EPA and DHA as a separation in a three-zone SMB system. They used C18 as the stationary phase and ethanol and alkaline water as the mobile phases. In a single-column experiment, the effect of the motional phase ratio, pH, and temperature on the separation in the SMB separation cell was investigated. Based on the obtained non-linear adsorption isotherms, a fully separated region was found according to triangular theory. The effect of sample concentration, flow ratio in the adsorption and fractionation zones, and column distribution on separation was further investigated. The final experimental results showed that the recovery and purity of EPA and DHA exceeded 99%, with a productivity of 4.15 g/L/h and a solvent consumption of 1.11 L/g [140].

Unlike the continuous operation of SMB, Li et al. employed a continuous intermittent chromatography system that combined intermittent chromatography with a continuous chromatographic mode of operation (called continuous intermittent chromatography). Under the continuous intermittent chromatography method, a self-created proprietary high-pressure multiport rotary valve is employed to elute target chemicals while simultaneously removing light and heavy contaminants. According to experimental findings, EPA ethyl ester has a relative maximum purity of 98.98% following continuous intermittent chromatography. The productivity of EPA ethyl ester under continuous intermittent chromatography was 5.48 times more that under intermittent chromatography, and the solvent consumption by intermittent chromatography was 1.27 times greater than that by continuous intermittent chromatography [141].

Simulated moving-bed chromatography involves inherently solid–liquid adsorption, which inevitably consumes a large number of stationary phase materials; however, the use of countercurrent chromatography addresses this problem. Counterflow chromatography is a liquid–liquid partitioning chromatography technique that uses two immiscible liquid phases to separate solutes based on different partition coefficients. The diverse choice of two-phase solvent systems avoids permanent adsorption of the target by eliminating the use of solid stationary phases. pH-zone-refining countercurrent chromatography (CCC) is a powerful and unique technique to separate weak acidic and basic compounds based on the differences in pKa values and hydrophobicity. Li et al. performed the first successful separation of free EPA and DHA using pH-zone-refined CCC. They investigated the effect of different solvent systems and concentrations of retention and eluant on the separation efficiency. The two-phase solvent system chosen was n-heptane/methanol/water (100:55:45, *v*/*v*), in which trifluoroacetic acid was used as the retention agent in the organic phase and ammonium hydroxide as the eluent in the aqueous phase. The purity of EPA was measured by gas chromatography as 95.5%, and the amplified separation of 1 g sample was also achieved successfully, yielding satisfactory yields and target purity [142].

In contrast to the previously described pH-zone-refined countercurrent chromatography, Fan et al. developed a novel two-phase system based on guanidine ionic liquids by replacing water with ionic liquids. The separation of EPA-EE and DHA-EE from commercial fish oil was achieved by a non-aqueous solvent system (n-heptane/methanol/ionic liquid, 1:1:5%, *v*/*v*/*m*). The alkaline solvent system, ionic liquid type, alkyl chain length, ionic liquid content, sample loading, preparative separation, and ionic liquid recovery were investigated and optimized. The final result allowed for high-purity EPA-EE and DHA-EE to be produced simultaneously in one cycle with a purity of 96.92% and 95.12%, respectively [143].

Compared with other methods, enzymatic treatment is considered an excellent choice for the EPA-EE monomer preparation. For example, EPA-EE is currently produced using liquid chromatography with nearly 100% purity, but the massive solvent and equipment energy consumption of this method has limited its application in industry. However, the advantage of enzymatic treatment is that the purity of the EPA-EE monomer reach nearly 100% with minimal solvent and energy consumption [116]. Therefore, enzymatic treatment can be considered the best extraction method for the isolation of EPA-EE monomers.

## 4. Subsequent EPA-EE Use and Its Storage Characteristics

### 4.1. EPA-EE Bioavailability

The major fish oil products available on the market are ethyl ester, glycerol-triglyceride, and free fatty acids. The bioavailability of any nutrient element is directly related to its digestive and absorption pathways. After oral ingestion of EPA-EE, fish oil is transformed into small, fine droplets due to the mechanical effects of gastric peristalsis. EPA-EE then binds to bile and phospholipids for efficient emulsification. Without bile and phospholipids, EPA-EE is not absorbed. The emulsified EPA-EE is cleaved into free eicosapentaenoic acid and ethanol molecules by pancreatic lipase in the small intestine. Subsequently, FFA is re-esterified to TAG or absorbed in the free form before being incorporated into celiac particles [144]. EPA-EE seems to be poorly activated by pancreatic lipase, and the inclusion of EPA-EE within the micellar structure appears to be limited. Therefore, the in vitro digestibility of EPA-EE is lower than that of the triglyceride type [145].

The currently accepted conclusion is that ethyl ester EPA has the lowest bioavailability; however, it cannot be denied that EPA-EE does achieve the highest EPA concentration. In pharmacology, bioavailability is a subset of absorption. It is defined as the portion of a drug’s supplied dose that enters the systemic circulation and is thus thought to be available for physiological effects on tissue. The bioavailability of nutrients follows the same fundamental principles, and relative bioavailability is typically evaluated by comparing the increase in an area under the curve following the consumption of a supplement product to a typical reference product [146]. This method of evaluation will appear in the following examples. From another perspective, it encompasses both bioaccessibility and bioactivity and provides a more comprehensive overview of a substance’s rate and amount of absorption [147]. However, bioavailability in research needs to be considered in a broader context, including with respect to the amount of substance that reaches the body’s circulation or where it performs its physiological activity [148]. We have typically considered animal studies and human trials in previous experimental studies. Measuring EPA concentrations in plasma, serum, blood cells, and lymphatic fluid is a more common measure of bioavailability. Plasma fatty acid levels represent the short-term or medium-long-term availability of fatty acids in the diet. Concentrations in blood cells are most often a good index of long-term bioavailability [148].

Many researchers suggest consuming formulae containing ethyl ester with high-fat foods, thereby increasing pancreatic lipase activity and facilitating EPA-EE absorption [92]. However, not everyone is suited to a high-fat diet; therefore, it is necessary to optimize EPA-EE absorption in those who consume a low-fat or fat-free diet. Researchers now aim to increase the bioavailability of ethyl ester formulations to achieve optimal absorption and utilization of ethyl esters under fasting conditions. Advanced lipid technologies (ALT) [149] and the self-microemulsifying delivery systems (SMEDSs) are two new technologies that have been created to improve EPA-EE absorption. Lopez-Toledano et al. developed an ethyl ester formulation based on ALT technology that was less affected by food effects and formed stable micelles in situ in the presence of an aqueous medium, independent of bile salt secretion [150]. Another case is the new ethyl ester formulation PRF-021. This self-microemulsifying delivery system improved the absorption of a single dose of EPA-EE under fasting conditions after the relative bioavailability of blood samples from the test population was measured [151]. A randomized double-blind study by Bremmell et al. also demonstrated that formulating EPA-EE with SEDSS concentrate (AquaCelle^®^) showed a significant improvement in oral absorption with no need for a high-fat meal [152].

Admittedly, the bioavailability of EPA-EE made for artificial purposes is lower than that of EPA in its natural form, but if it can be encapsulated in a food-grade delivery system or retaken by other means, then EPA-EE can be brought to a greater level of functional activity. Whereas numerous studies have concentrated on the bioavailability of EPA and DHA, the mechanism of their uptake into absorptive epithelial cells and intracellular events leading to basolateral secretion still depend on the understanding of long-chain fatty acids [153]. It is more difficult to study bioavailability. The intake of PUFA decreases under some pathological conditions, such as malabsorption. Its chemical form most greatly influences the bioavailability of EPA. It is also affected by the way it is consumed, how meals interact with one another, how much peroxidation occurs, and other factors [92,146]. Furthermore, the form of intake can be interpreted as the form in which it is stored, e.g., whether it is a capsule or a tablet, etc. These factors will be discussed in the next section.

### 4.2. EPA-EE Storage Properties and Oxidative Stability

As a functional lipid in fish oil, eicosapentaenoic acid contains five carbon–carbon double bonds, which renders fish oil very prone to lipid oxidation, producing toxic and harmful oxidation products and leading to a deterioration of its quality and loss of nutritional value—or worse, endangering the health of those who take it [154]. Fish oils and ethyl esters are more susceptible to oxidation than triglycerides, so many challenges need to be tackled before they can be incorporated into commercially available functional foods. The oxidative stability of fish oil is intimately tied to its storage form. It might be argued that every storage type is developed to improve bioavailability and oxidative stability.

Unlike other edible oils, fish oils rich in PUFAs are more likely to trigger lipid oxidation and subsequent destabilization, especially in the form of ethyl esters rather than triglycerides. Therefore, oxidative stability is the determining factor in the use of EPA-EE in its pure form or as a nutritional fortification in food [28]. Understanding the different oxidation mechanisms and factors affecting the oxidation dynamics responsible for enhancing oxidative stability in EPA-EE fish oil is crucial and is also an area of interest. Lipid oxidation mechanisms comprise autoxidation, photosensitive oxidation [28], and enzymatic oxidation [155]. Fatty acid composition is an influential element in lipid oxidation, as higher levels of eicosapentaenoic acid in fish oils lead to higher oxidative susceptibility. The resulting number of diallyl methylene in PUFAs is a characteristic structural feature that makes fish oils more susceptible to oxidation [28]. Another key influence on lipid oxidation is the storage temperature; higher storage temperatures are more likely to lead to lipid oxidation and hence to increased accumulation of oxidation byproducts. However, lower storage temperatures can only help slow lipid oxidation but not stop the process completely [156]. Factors affecting lipid oxidation are complex and multifaceted. In addition to the abovementioned factors, manipulation during processing, the addition of antioxidants, and post-encapsulation storage forms may cause lipid oxidation, especially in obtaining high-purity EPA-EE fish oil formulations.

Researchers have developed different strategies to prevent or retard oxidation. Because oxygen plays an important role lipid oxidation, many measures can be taken to control the oxygen content in fish oil, such as through the use of deoxygenated active packaging, which can greatly prolong its oxidative stability and improve the nutritional quality of food without additional of exogenous antioxidants [157]. Another simpler choice is to add antioxidants, such as tocopherols, phenolic compounds, ascorbate, and synthetic antioxidants, which are widely used in foods to prevent lipid oxidation in all sorts of products [158]. While combining omega-3 fatty acids by esterification with natural oxidants via the action of lipases would be a novel option, Dey et al. evaluated the ability of lutein and omega-3 PUFAs, both of which are prone to autoxidation, to be stabilized by esterification. The final results showed that the esterified form significantly facilitated the protective effect of oxidative degradation of bulk fish oil, resulting in fish oil free from secondary oxidation and thus extending its shelf life [159]. Not coincidentally, Liu et al. also applied the esterification of EPA with quercetin to enhance oxidative stability. Compared with the quercetin parent, the new product showed much higher lipophilicity, resulting in increased cell membrane affinity; thus, the antioxidant activity of the cells was enhanced. However, further studies are needed to determine whether it improved the biological activity [160]. The omega-3 phenolics produced by structural modification that were used to fight the oxidation problem of fish oil can be introduced as a high-nutritional-value food products [28].

In addition to using effective antioxidants or new forms of esterification, there has been ongoing interest in using food systems to encapsulate, protect, and deliver EPA-rich fish oils with improved oxidative stability. Encapsulation and microencapsulation represent two excellent candidates.

According to the common theory of free radical oxidation, EPA should be oxidized quickly because of the high amount of diallyl hydrogen in the molecule. However, some experimental results indicate that these polyunsaturated fatty acids have unusually high oxidative stability in biological systems and certain foods [161]. Current delivery systems for encapsulating Omega-3 fish oils include liposomes, solid lipid nanoparticles (SLN), nanostructured lipid carriers (NLC), complex multiple-emulsion systems, microgels, nanofibers, and inclusion complexes, among other forms [162]. Fish oil encapsulating techniques that are extensively used include oil-in-water emulsions and nanoemulsions. The bioavailability and physical stability of formulations containing omega-3 fish oil can be greatly enhanced by the smaller size of droplets in nanoemulsions [163]. Emulsions and nanoemulsions are typically formed by homogenizing the oil and water phases in the presence of an emulsifier. SLNs and NLCs can be formed using methods similar to those used to make emulsions or nanoemulsions but with homogenization performed above the melting point of the lipid phase. The formulation is then cooled below the crystallization temperature to stimulate a liquid-to-solid transition in the lipid phase [164]. Multinanometer emulsions have a more complex structure than conventional emulsions. Water–oil–water (W/O/W) is best-suited for encapsulating fish oils, allowing for controlled release and protection of sensitive ingredients [165]. Unlike the emulsion form described above, liposomes are more like “clear liquids”. Liposomes typically consist of phospholipids organized into a bilayer structure, with fish oils interspersed between the hydrophobic tails of the phospholipids. Nanoliposomes exhibit higher encapsulation rates and oxidative stability when using innovative methods such as microfluidics and ultrasound [166]. However, liposomes are difficult to produce on a large scale and have low physical stability in complex food matrices. Today, the most commercially functional fish oil is packaged in gelatin capsules. Therefore, embedding oil droplets into gelling biopolymers is a worthwhile approach, such as by encapsulating fish oil into hydroxypropyl methylcellulose-maltodextrin and whey protein-gum arabic [165]. A similar method was used to capture bioactive substances into cyclic dextrins to form molecular inclusion complexes, which improved the oxidative stability of fish oil after the formation of complexes [167]. Nanofibers are made up of long, thin, fibrous materials typically assembled from food-grade biopolymers for encapsulation and controlled release of hydrophobic substances, as well as for improved oxidative stability, such as using electrospraying methods to encapsulate fish oil in zein nanofibers [168].

With the advanced encapsulation techniques described above, further applications of microencapsulation can also facilitate improved storage and stability. For the conversion of fluid fish oil formulations or emulsions into powder form, it is not only the technique used to achieve drying that needs to be considered but also the choice of wall material. The choice of wall material is important to achieve the final microencapsulation. It has a key role in the functional properties of the formed powder (flowability, packaging, encapsulation efficiency, and chemical stability). Commonly used wall materials can be divided into carbohydrates and proteins; the amount of wall materials varies, and the encapsulation rate shows different results [162]. Now, it is more common to use spray drying, which is economical and can be performed on an industrial scale [169]. Another option is freeze drying, which is more costly and time-consuming and yields less. Its advantage is the reduced susceptibility to oxidation [170]. Lastly, fluid materials may also be converted to powder using electrospray or electrostatic spinning techniques [168].

Regardless of the perspective, the oxidative stability and bioavailability of ethyl fish oil are crucial. The oxidative stability of EPA-EE in its pure form is weaker than that of triglycerides and phospholipid-based eicosapentaenoic acid at high concentrations. Therefore, for more efficient utilization of EPA-EE by the human body, while avoiding the harmful effects of nutritional value and biological function deterioration due to lipid peroxidation, higher bioavailability and oxidative stabilization are required with appropriate food delivery systems and antioxidant methods. In addition to the use of stabilization techniques to combat fish oil oxidation mentioned above, the synergistic effect of antioxidant substances and active packaging provides potential rationale the use for fish oil mass oxidation [171]. Additional consideration and in-depth study are also needed to determine whether newly created stabilizing approaches, such as structural alterations and antioxidant inclusion into newly generated compounds, are safe for human intake. Consequently, stabilization techniques applied to omega-3-enriched fish oils remain an important and hot topic for continued investigation and research by academia and industry. Figure 4 shows the most recent techniques used to increase the bioavailability and the primary storage strategies for EPA-EE (or highly concentrated omega-3 PUFAs).

## 5. Conclusions and Future Prospects

Due to the benefits to health and the challenge of consuming enough omega-3 PUFAs to meet human needs, there was initially a considerable market demand for Omega-3 PUFA concentrates. The health advantages of a combination of omega-3 PUFAs and the advantages of DHA for the brain attracted more focus in earlier studies [7,172]. It would be logical and the right decision to separate EPA- and DHA-rich concentrates and to prepare monomers because there are many functional differences between EPA and DHA, and these differences show that EPA has a more significant role than DHA in, for example, major depression, heart disease, and cardiovascular disease.

A longer-term strategy is necessary, frequently necessitating the merging of several areas, to enable greener and more affordable access to high-value EPA-EE monomers. Fish oil can be made from microalgae and microorganisms, in addition to actual fish [173,174]. This novel approach can enhance the production of EPA or DHA and even reduce the cost of post enrichment. Likewise, adjusting the ratio of omega-3 PUFAs in farmed fish feeds can enhance the EPA and DHA content of crude fish oil extracts in another way [175,176]. For a more environmentally sustainable use of EPA and DHA, microalgae and genetically modified crops are highly promising as sources for the development of these functional fatty acids [175]. Considering the process of extracting, refining, and concentrating fish oil, it is important to use alternative green technologies and consider their economic cost. With the aid of certain protein engineering modifications of natural lipases or more appropriate immobilization tools to increase the number of enzymes used, lipid extraction and EPA enrichment can be maximized, and even ethyl esterification and high purity of fish oil can be achieved in one step [84,120]. The development and use of enzyme tools are promising and interesting areas. The use of some emerging green solvents to replace the common organic solvents that are toxic and harmful to humans is also meaningful. However, an economically circular solution needs to be designed [59,60]. Additionally, with the improvement of enzyme activity and the increase in the number of cycles of enzyme use, it is expected that the enzymatic method will be an excellent choice to obtain EPA-EE concentrate in the future. Moreover, studies of the processes involved in the formation of EPA-EE should not be limited to laboratory models but involve the use of factories and large-scale equipment, such as simulated moving bed chromatography, enabling the evaluation of its energy efficiency and economic feasibility [177].

The limited bioavailability and low oxidative stability of EPA-EE have always been challenging problems. In addition of the ability to control by the addition of synthetic antioxidants, enzyme-catalyzed esterification to form new esters is a promising approach, although its bioactivity and bioavailability require further evaluation [160]. The protection of polyunsaturated fatty acids against autoxidation brought on by environmental factors using food system encapsulation and microencapsulation is currently a realistic and effective technology, boosting their oxidative stability. This can improve the taste and acceptability of fish oil supplements, in addition to substantially reducing the smell of fish oil. Self-emulsification can also increase the bioavailability at low fat levels, although this must be considered in terms of the financial cost. Future research attempting to metabolically engineer the possibility of increasing the levels of natural antioxidants in PUFA-rich oils may be able to get to the root of the problem. Moreover, the development of packaging with antioxidant activity may also be a promising approach [171]. Special attention should be paid to the need for more studies on the hazards of peroxides during the production, use, and storage of EPA-EE.

## Figures and Tables

**Figure 1 molecules-28-00672-f001:**
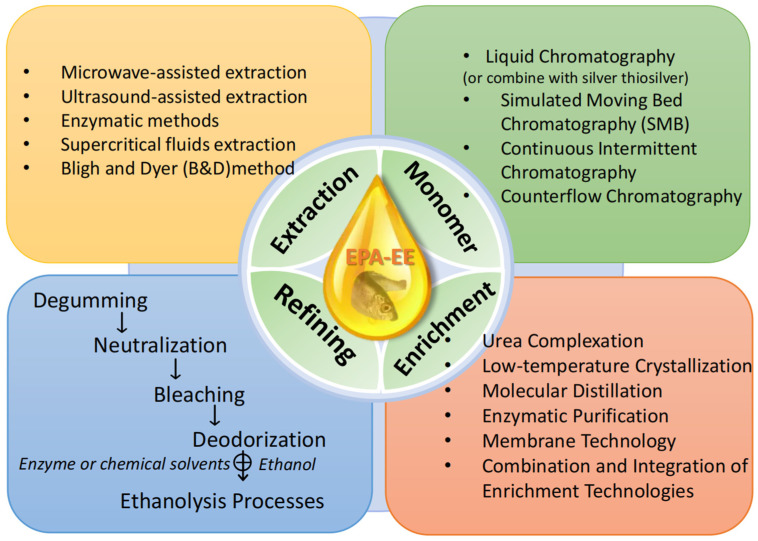
An overview of the main processing steps and the latest technology in the EPA-EE formation process.

**Figure 2 molecules-28-00672-f002:**
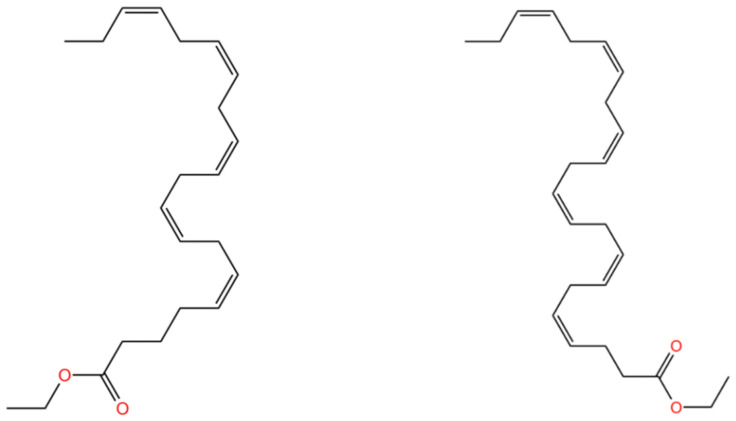
The chemical structural formula of EPA-EE (**left**) and DHA-EE (**right**).

**Figure 3 molecules-28-00672-f003:**
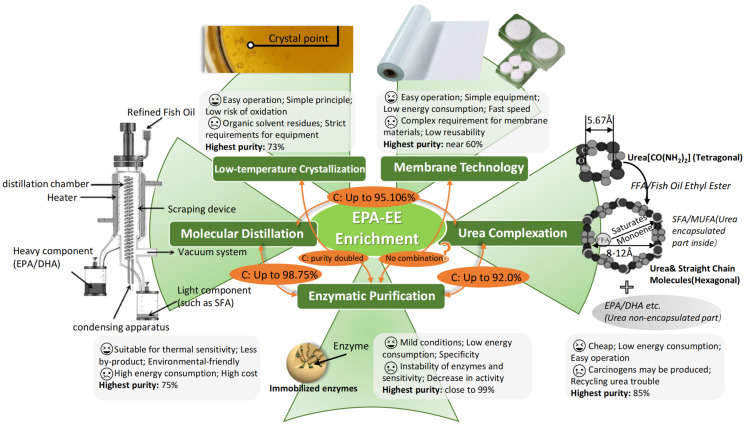
An overview of the features of EPA-EE enrichment technologies and the results of the combination of these technologies. C, combination; the orange double-headed arrow indicates a combination of methods; highest purity generally means the highest concentration achieved by the EPA-EE mixture; smiley faces indicate the advantages of the technology, and sad faces indicate disadvantages.

**Figure 4 molecules-28-00672-f004:**
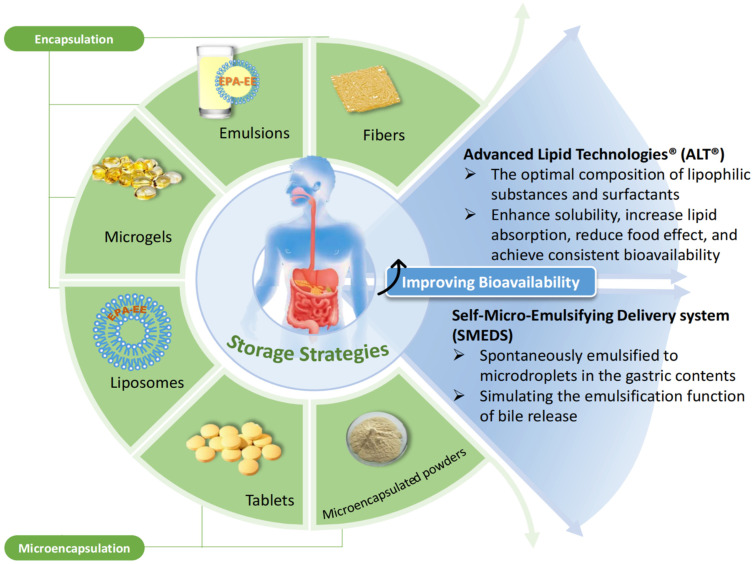
The most recent techniques to increase bioavailability, as well as the primary storage forms of EPA-EE (or highly concentrated omega-3 PUFAs).

**Table 3 molecules-28-00672-t003:** Comparison of fish oil extraction methods.

Extraction Method	Principle	Advantages	Drawbacks	References
Microwave-Assisted Extraction (MAE)	Microwave energy is used to rapidly heat a solid sample in contact with a solvent	High extraction rates; reduced extraction time and solvent consumption	High power consumption; difficulty in mass production; risk of oxidation	[39,40,41,42]
Ultrasound-Assisted Extraction (UAE)	Ultrasound is used to penetrate the solvent in contact with the lipid and thus enhance solvent penetration	High extraction rates; reduced extraction time and solvent consumption	Difficult to scale up; high power consumption	[43,44,45,46]
Enzymatic Methods	Enzymatic specificity	Requires no organic solvent; low energy consumption	High price; difficult to mass-produce; reduction in enzyme activity and enzyme recovery	[47,48,49]
Supercritical Fluid Extraction (SFE)	SC-CO_2_ is used as the solvent	Fast, efficient, and highly productive; no organic solvent needed; high purity; low-temperature operation	Requires expensive and complex equipment; high power consumption	[50,51,52]

**Table 4 molecules-28-00672-t004:** Properties of EPA-EE and DHA-EE.

Substance	Molecular Weight	Melting Point	Flash Point	Boiling Point	Intensity	Refractive Index	1-Octanol/Water Partition Coefficient (log P)	References
EPA-EE	330.50	-	103.1 ± 24.0	417.0 ± 34.0	0.909 ± 0.06	-	7.642 ± 0.362	[93]
DHA-EE	356.54	-	102.1 ± 21.2	443.5 ± 24.0	0.914 ± 0.06	-	8.154 ± 0.375	

**Table 5 molecules-28-00672-t005:** Comparison of different concentration processes.

	Urea Complexation	Molecular Distillation	Low-Temperature Crystallization	Enzymatic Purification	Membrane Technology	Liquid Chromatography	Supercritical Fluid Chromatography	Supercritical Fluid Fractionation
Enrichment Mechanism	Degree of saturation	Boiling point	Melting point	Enzyme selection specificity	Pore size and chemical affinity	Chain length and degree of unsaturation	Chain length and degree of unsaturation	Chain length
Conditions	−10–90 °C, 1 bar	140–220 °C, 0.001 mbar	−70–0 °C, 1 bar	25–65 °C, 1 bar	25–40 °C, 3–6 bar	20–50 °C, 1 bar	35–50 °C, >140 bar	35–50 °C, >140 bar
Omega-3 purity	60–99%	65–75%	60–90%	46–99%	35–54%	>90%	>90%	75–85%
Operation Mode	Batch	Continuous	Batch	Batch or semi-batch	Batch	Semi-batch	Continuous	Continuous
Risk of Oxidation	Possible	Low	Possible	Low	Low	Possible	Low	Low
Capital Investment	Low	Moderate	Moderate	Moderate	Moderate	High	High	High
Reference	[95,96,99]	[94,112]	[106,107,108,109]	[114,116,117,118,119]	[122,123,124,125]	[129]	[80]	[80]

## Data Availability

Not applicable.

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
