# Peer review of "Highly Valuable Fish Oil: Formation Process, Enrichment, Subsequent Utilization, and Storage of Eicosapentaenoic Acid Ethyl Esters"

_molecules, 2023, doi:10.3390/molecules28020672_

Round 1

Reviewer 1 Report

Dear Authors,

Demand for significantly expand the scientific knowledge of high-purity eicosapentaenoic acid ethyl ester (EPA-EE) properties, which more efficiently than mixed Omega-3 polyunsaturated fatty acid preparations in diseases such as hyper-lipidemia, heart disease, major depression, and heart disease, hence the market demand for EPA-EE is in great demand today. Thus, because of this, writing a review article is also highly desirable. Unfortunately, the manuscript presented for review contains only very general descriptions of the methods of separation and analysis, and these must be presented very precisely in many analytical variants. The reviewed article does not contain this, and therefore it looks like propaganda material for some newspaper or medical advertising brochure. For this reason, it is not suitable for printing, although, as I must emphasize, the subject considered here is extremely important for humanity.

Author Response

Response to Editor and Reviewers

Molecules-2063710

Manuscript title: High Valuable Fish Oil: Formation process, Enrichment, Subsequent Utilization, and Storage of Eicosapentaenoic Acid Ethyl Esters

Dear Sir,

    Thanks very much for your email and comments as well as the reviewers’ comments on the manuscript. These comments are all valuable and very helpful to revise and improve our manuscript. We are pleased to answer all the questions of reviewers and the manuscript has also been revised according to the comments point by point. Revised portions have been changed with track changes in the revised manuscript. The following summarizes the main revisions made in the manuscript.

Response to Editor and Reviewer 1 Comments:

Reviewer #1: Demand for significantly expand the scientific knowledge of high-purity eicosapentaenoic acid ethyl ester (EPA-EE) properties, which more efficiently than mixed Omega-3 polyunsaturated fatty acid preparations in diseases such as hyper-lipidemia, heart disease, major depression, and heart disease, hence the market demand for EPA-EE is in great demand today.

1) Thus, because of this, writing a review article is also highly desirable. Unfortunately, the manuscript presented for review contains only very general descriptions of the methods of separation and analysis, and these must be presented very precisely in many analytical variants. The reviewed article does not contain this, and therefore it looks like propaganda material for some newspaper or medical advertising brochure. For this reason, it is not suitable for printing, although, as I must emphasize, the subject considered here is extremely important for humanity.

Response: Thanks for your comment. In order to present the analytical variants precisely, correlations have been made based on your suggestions.

To more accurately depict the variations and links among the various extract methods, the part of fish oil extraction has been clarified based on your suggestions (L111-L132; L137-L142; L146-L161; L167-L177; L211-L213; L215-L216; L195-L202; L219-L222).

In order to more effectively assess the influence of the various processes on the outcomes, the fish oil refining procedure as well as ethyl esterification described in the manuscript have been clarified based on your suggestions (L258-L275; L370-L372).

In addition, the analyses of the various methods and information related to the subsequent utilization of ethyl fish oil also have been added based on your suggestions. (L451-L458; L499-L515; L603-L607; L713-L722; L730-L735; L744-L755; L839-L846; L901-L908; L917-L921; L1022-L1024).

Finally, the language of this manuscript has been polished by native English speakers, and I hope it could satisfy the editor and you. Thanks again.

We believe that the revisions made based on those comments have significantly improved the manuscript. We look forward to your information about our revised manuscript and thank you again.

Best regards,

Dr Gangcheng Wu

Reviewer 2 Report

The article “High Valuable Fish Oil: Formation process, Enrichment, Subsequent Utilization, and Storage of Eicosapentaenoic Acid Ethyl Esters” is a very interesting review article that demonstrates how EPA-EE is manufactured with a process that involves the extraction, refining, and methanolysis of EPA-EE. This manuscript is well-written. However, a few minor points need to be addressed:

Comments:

Is there a difference in the purity ratio of EPA-EE monomers by different methods?

Do the authors care to speculate on the best extraction methods in isolation of EPA-EE monomers?

Maybe, the authors could mention the limitations of each technique.

Author Response

Response to Editor and Reviewers

Molecules-2063710

Manuscript title: High Valuable Fish Oil: Formation process, Enrichment, Subsequent Utilization, and Storage of Eicosapentaenoic Acid Ethyl Esters

Dear Sir,

    Thanks very much for your email and comments as well as the reviewers’ comments on the manuscript. These comments are all valuable and very helpful to revise and improve our manuscript. We are pleased to answer all the questions of reviewers and the manuscript has also been revised according to the comments point by point. Revised portions have been changed with track changes in the revised manuscript. The following summarizes the main revisions made in the manuscript.

Response to Editor and Reviewer2 Comments:

Reviewer #2: The article “High Valuable Fish Oil: Formation process, Enrichment, Subsequent Utilization, and Storage of Eicosapentaenoic Acid Ethyl Esters” is a very interesting review article that demonstrates how EPA-EE is manufactured with a process that involves the extraction, refining, and methanolysis of EPA-EE. This manuscript is well-written. However, a few minor points need to be addressed:

Comments:

1) Is there a difference in the purity ratio of EPA-EE monomers by different methods?

Response: Thanks for your comment, and the answer is yes. We are sorry that the manuscript did not clarify it well. We have added a precise description of the purity of EPA-EE in the different methods based on your suggestion (L437-L439; L442-L444; L453-L454; L502-L503; L513-L515; L579-L580; L586-L588; L657-L658; L688-L689; L734-L735; L741-L743; L766-L769; L788-L790; L807-L808; L827-L829; L836-L838).

2) Do the authors care to speculate on the best extraction methods in isolation of EPA-EE monomers?

Response: Thanks for your comment. Compared with other methods, the enzymatic treatment has been considered as the excellent choice for the EPA-EE monomer preparation. For example, EPA-EE is currently produced using liquid chromatography with nearly 100%, but the massive solvent and equipment energy consumption of this method has limited its application in the industry. Whereas, the advantage of enzymatic treatment is the purity of EPA-EE monomer could reach nearly 100% with minimal solvent and consuming less energy. Therefore, the enzymatic treatment could be considered as the best extraction methods in isolation of EPA-EE monomers. The correlation has been made as suggested (L839-L846).

References:

Liu, Y.; Dave, D. Recent progress on immobilization technology in enzymatic conversion of marine by-products to concentrated omega-3 fatty acids. Green Chem. 2022, 24, 1049-1066, doi:10.1039/d1gc03127a.

Gao, K.; Chu, W.; Sun, J.; Mao, X. Identification of an alkaline lipase capable of better enrichment of EPA than DHA due to fatty acids selectivity and regioselectivity. Food Chem. 2020, 330, 127225, doi:10.1016/j.foodchem.2020.127225.

Moreno-Perez, S.; Turati, D.F.; Borges, J.P.; Luna, P.; Senorans, F.J.; Guisan, J.M.; Fernandez-Lorente, G. Critical Role of Different Immobilized Biocatalysts of a Given Lipase in the Selective Ethanolysis of Sardine Oil. J Agric Food Chem 2017, 65, 117-122, doi:10.1021/acs.jafc.6b05243.

Akanbi, T.O.; Adcock, J.L.; Barrow, C.J. Selective concentration of EPA and DHA using Thermomyces lanuginosus lipase is due to fatty acid selectivity and not regioselectivity. Food Chem. 2013, 138, 615-620, doi:10.1016/j.foodchem.2012.11.007.

3) Maybe, the authors could mention the limitations of each technique.

Response: Thanks for your comment. The correlation has been made as suggested (L143 Table 3; L468-L470; L481-L482; L519-L521; L554-L556; L620-L623; L672-L673; L841-L843).

We believe that the revisions made based on those comments have significantly improved the manuscript. We look forward to your information about our revised manuscript and thank you again.

Best regards,

Dr Gangcheng Wu

Reviewer 3 Report

The manuscript entitled High Valuable Fish Oil: Formation process, Enrichment, Subsequent Utilization, and Storage of Eicosapentaenoic Acid Ethyl Esters has been peer-reviewed.

The work is scientifically sound and has a good literature review. The manuscript is well-organized. However, I have noted some drawbacks.

1. The graphical abstract (G.A.) does not summarizes the manuscript content. Perhaps, you could mix the G.A. and Fig.2 into a new G.A.

2. I missed the use of more figures or fluxograms in each topic, (despite the graphical abstract and Fig. 2) to improve the understanding of it.

3. In the item 2.1. Fish Oil Extraction, you could add information (organized in a new table) regarding the major sources (fish species) to oil extraction, production amount, major producing countries, oil yeild, and extraction type.

4. Please, check though all text for scientific names that are without the italic highlight (i.e. P. fluorescens).

5. In Table 2 phrases should begin with capital letters.

6. Table 3 should have references.

Author Response

Response to Editor and Reviewers

Molecules-2063710

Manuscript title: High Valuable Fish Oil: Formation process, Enrichment, Subsequent Utilization, and Storage of Eicosapentaenoic Acid Ethyl Esters

Dear Sir,

    Thanks very much for your email and comments as well as the reviewers’ comments on the manuscript. These comments are all valuable and very helpful to revise and improve our manuscript. We are pleased to answer all the questions of reviewers and the manuscript has also been revised according to the comments point by point. Revised portions have been changed with track changes in the revised manuscript. The following summarizes the main revisions made in the manuscript.

Response to Editor and Reviewer3 Comments:

Reviewer #3: The manuscript entitled High Valuable Fish Oil: Formation process, Enrichment, Subsequent Utilization, and Storage of Eicosapentaenoic Acid Ethyl Esters has been peer-reviewed.The work is scientifically sound and has a good literature review. The manuscript is well-organized. However, I have noted some drawbacks.

1) The graphical abstract (G.A.) does not summarizes the manuscript content. Perhaps, you could mix the G.A. and Fig.2 into a new G.A.

Response: Thanks for your comment. The new G.A. has been clarified in the revised manuscript (Fig 1).

2) I missed the use of more figures or fluxograms in each topic, (despite the graphical abstract and Fig. 2) to improve the understanding of it.

Response: Thanks for your comment. This information has been added in the revised manuscript (Fig 2 and other figures).

3) In the item 2.1. Fish Oil Extraction, you could add information (organized in a new table) regarding the major sources (fish species) to oil extraction, production amount, major producing countries, oil yeild, and extraction type.

Response: Thanks for your comment. This information was added to the revised manuscript based on your suggestions (L134-L135, Table 2).

4) Please, check though all text for scientific names that are without the italic highlight (i.e. P. fluorescens).

Response: Thanks for your comment. This information was added to the revised manuscript based on your suggestions (L191; L344; L566; L582; L731-732).

5) In Table 2 phrases should begin with capital letters.

Response: Thanks for your comment. Corrections have been made as suggested (L135, Table 3, the original Table 2).

6) Table 3 should have references.

Response: Thanks for your comment. Corrections have been made as suggested (L400, Table 4, the original Table 3).

We believe that the revisions made based on those comments have significantly improved the manuscript. We look forward to your information about our revised manuscript and thank you again.

Best regards,

Dr Gangcheng Wu

Round 2

Reviewer 1 Report

Dear Authors,

After the suggested corrections, the present form of the manuscript is better than the previous one, however:

-        Lines 111-117 – are the Authors sure that these sentences are logical, i.e. “…as raw materials to make fish oil in the 19th century by crushing and grinding the cooked fish. - When boiling at high temperatures, the long-chain unsaturated fatty acids are also vulnerable to oxidation and degradation.”(?),

-        Lines 134-136 – adding the Table2 is absolutely justified, but it must be carefully interpreted(!),

-        Line 148 – the term “Folch technique” is mentioned only once in the text without a proper reference and therefore needs to be explained in detail,

-        Lines 261-264 – it is essential to explain why “the extraction of free fatty acids with betaine monohydrate stabilizes both the antioxidant content and the extraction process”(?). The lack of such an explanation makes the text laconic(!),

-        Lines 620-623 – this is a long sentence, the correct meaning of which is complicated to interpret unambiguously and does not explain why “…the fish oil ethyl esterification process will be accelerated”(?).

-        I passionately believe that after the corrections, the section "Conclusions" should be adapted to the current content of the manuscript!

Author Response

Response to Editor and Reviewers

Molecules-2063710

Manuscript title: High Valuable Fish Oil: Formation process, Enrichment, Subsequent Utilization, and Storage of Eicosapentaenoic Acid Ethyl Esters

Dear Sir,

    Thanks very much for your email and comments as well as the reviewers’ comments on the manuscript. These comments are all valuable and very helpful to revise and improve our manuscript. We are pleased to answer all the questions of reviewers and the manuscript has also been revised according to the comments point by point. Revised portions have been changed with track changes in the revised manuscript. The following summarizes the main revisions made in the manuscript.

Response to Editor and Reviewer 1 Comments:

Reviewer #1: After the suggested corrections, the present form of the manuscript is better than the previous one, however:

1) Lines 111-117 – are the Authors sure that these sentences are logical, i.e. “…as raw materials to make fish oil in the 19th century by crushing and grinding the cooked fish. - When boiling at high temperatures, the long-chain unsaturated fatty acids are also vulnerable to oxidation and degradation.”(?),

Response: Thanks for your comment. Corrections have been made as suggested(L116-L124).

References:

Rubio-Rodríguez, N.; Beltrán, S.; Jaime, I.; de Diego, S.M.; Sanz, M.T.; Carballido, J.R. Production of omega-3 polyunsaturated fatty acid concentrates: A review. Innov. Food Sci. Emerg. Technol. 2010, 11, 1-12, doi:10.1016/j.ifset.2009.10.006.

Wang, J.; Han, L.; Wang, D.; Sun, Y.; Huang, J.; Shahidi, F. Stability and stabilization of omega-3 oils: A review. Trends Food Sci. Technol. 2021, 118, 17-35, doi:10.1016/j.tifs.2021.09.018.

2) Lines 134-136 – adding the Table2 is absolutely justified, but it must be carefully interpreted(!),

Response: Thanks for your comment. This information has been added in the revised manuscript (L109-L116).

References:

Gladyshev, M.I.; Sushchik, N.N.; Tolomeev, A.P.; Dgebuadze, Y.Y. Meta-analysis of factors associated with omega-3 fatty acid contents of wild fish. Reviews in Fish Biology and Fisheries 2017, 28, 277-299, doi:10.1007/s11160-017-9511-0.

Moxness Reksten, A.; Ho, Q.T.; Nostbakken, O.J.; Wik Markhus, M.; Kjellevold, M.; Bokevoll, A.; Hannisdal, R.; Froyland, L.; Madsen, L.; Dahl, L. Temporal variations in the nutrient content of Norwegian farmed Atlantic salmon (Salmo salar), 2005-2020. Food Chem. 2022, 373, 131445, doi:10.1016/j.foodchem.2021.131445.

3) Line 148 – the term “Folch technique” is mentioned only once in the text without a proper reference and therefore needs to be explained in detail,

Response: Thanks for your comment. Folch is the author’s name, and we did not cited this reference properly. This information has been added in the revised manuscript (L155-L157).

References:

FOLCH, J.; LEES, M.; STANLEY, G.H.S. A simple method for the isolation and purification of total lipids from animal tissues. The Journal of biological chemistry 1959, Vol.226, 497-509.

4) Lines 261-264 – it is essential to explain why “the extraction of free fatty acids with betaine monohydrate stabilizes both the antioxidant content and the extraction process”(?). The lack of such an explanation makes the text laconic(!),

Response: Thanks for your comment. Corrections have been made as suggested(L270-L278).

References:

Zahrina, I.; Nasikin, M.; Krisanti, E.; Mulia, K. Deacidification of palm oil using betaine monohydrate-based natural deep eutectic solvents. Food Chem. 2018, 240, 490-495, doi:10.1016/j.foodchem.2017.07.132.

5) Lines 620-623 – this is a long sentence, the correct meaning of which is complicated to interpret unambiguously and does not explain why “…the fish oil ethyl esterification process will be accelerated”(?).

Response: Thanks for your comment. Corrections have been made as suggested(L634-L639).

References:

Liu, Y.; Dave, D. Recent progress on immobilization technology in enzymatic conversion of marine by-products to concentrated omega-3 fatty acids. Green Chem. 2022, 24, 1049-1066, doi:10.1039/d1gc03127a.

6) I passionately believe that after the corrections, the section "Conclusions" should be adapted to the current content of the manuscript!

Response: Thanks for all your comments, and the current content of the manuscript has been improved based on your suggestions. Thanks again.

We believe that the revisions made based on those comments have significantly improved the manuscript. We look forward to your information about our revised manuscript and thank you again.

Best regards,

Dr Gangcheng Wu

Round 3

Reviewer 1 Report

Dear Authors,

The Authors ‘revised the original manuscript reasonably quickly’, but they did so in a very careless manner:

- in line 156, they added citation [177] non-chronologically, i.e. as the last one in the Bibliography, and at the same time, they placed citation [52] on the same page (see line 160). Is this supposed to be a joke?

- in the Bibliography, the Authors do not distinguish between the websites of the articles and the digital object identifier (i.e. DOI). For these reasons, the entire Bibliography is not edited correctly.

- the drawings are illegible and therefore need to be corrected!

Author Response

Response to Editor and Reviewers

Molecules-2063710

Manuscript title: High Valuable Fish Oil: Formation process, Enrichment, Subsequent Utilization, and Storage of Eicosapentaenoic Acid Ethyl Esters

Dear Sir,

    Thanks very much for your email and comments as well as the reviewers’ comments on the manuscript. These comments are all valuable and very helpful to revise and improve our manuscript. We are pleased to answer all the questions of reviewers and the manuscript has also been revised according to the comments point by point. Revised portions have been changed with track changes in the revised manuscript. The following summarizes the main revisions made in the manuscript.

Response to Editor and Reviewer 1 Comments:

Reviewer #1: The Authors ‘revised the original manuscript reasonably quickly’, but they did so in a very careless manner:

1)in line 156, they added citation [177] non-chronologically, i.e. as the last one in the Bibliography, and at the same time, they placed citation [52] on the same page (see line 160). Is this supposed to be a joke?

Response: Thanks for your comment. We apologize for making such mistakes, and the rest of the manuscript has also been carefully checked. Corrections have been made as suggested (L155-L156; L159-L160).

2)in the Bibliography, the Authors do not distinguish between the websites of the articles and the digital object identifier (i.e. DOI). For these reasons, the entire Bibliography is not edited correctly.

Response: Thanks for your comment. Corrections have been made as suggested (L1107; L1130; L1142; L1146; L1164; L1170; L1179; L1187-L1188; L1195; L1200-L1201; L1212; L1219; L1224; L1242; L1249; L1257; L1279; L1287; L1298; L1317-L1318; L1321; L1332; L1335; L1389; L1398; L1403-L1404; L1418; L1432; L1437; L1460; L1466; L1468; L1497; L1536; L1556; L1580; L1583; L1586; L1604; L1608; L1612; L1615; L1618-L1619; L1623; L1627; L1632; L1637; L1639-L1640).

3) the drawings are illegible and therefore need to be corrected!

Response: Thanks for your comment. This information has been changed in the revised manuscript (Fig 1 and other figures).

We believe that the revisions made based on those comments have significantly improved the manuscript. We look forward to your information about our revised manuscript and thank you again.

Best regards,

Dr Gangcheng Wu
